# Unsupervised Graph Neural Architecture Search with Disentangled Self-supervision

Zeyang Zhang[1]*, Xin Wang[1]†, Ziwei Zhang[1], Guangyao Shen[2], Shiqi Shen[2], Wenwu Zhu[1]†

[1]Department of Computer Science and Technology, BNRist, Tsinghua University, [2]Wechat, Tencent

zy-zhang20@mails.tsinghua.edu.cn, {xin_wang, zwzhang}@tsinghua.edu.cn,
{lucasgyshen, shiqishen}@tencent.com, wwzhu@tsinghua.edu.cn

## Abstract

The existing graph neural architecture search (GNAS) methods heavily rely on supervised labels during the search process, failing to handle ubiquitous scenarios where supervisions are not available. In this paper, we study the problem of unsupervised graph neural architecture search, which remains unexplored in the literature. The key problem is to discover the latent graph factors that drive the formation of graph data as well as the underlying relations between the factors and the optimal neural architectures. Handling this problem is challenging given that the latent graph factors together with architectures are highly entangled due to the nature of the graph and the complexity of the neural architecture search process. To address the challenge, we propose a novel Disentangled Self-supervised Graph Neural Architecture Search (**DSGAS**) model, which is able to discover the optimal architectures capturing various latent graph factors in a self-supervised fashion based on unlabeled graph data. Specifically, we first design a disentangled graph super-network capable of incorporating multiple architectures with factor-wise disentanglement, which are optimized simultaneously. Then, we estimate the performance of architectures under different factors by our proposed self-supervised training with joint architecture-graph disentanglement. Finally, we propose a contrastive search with architecture augmentations to discover architectures with factor-specific expertise. Extensive experiments on 11 real-world datasets demonstrate that the proposed **DSGAS** model is able to achieve state-of-the-art performance against several baseline methods in an unsupervised manner.

## 1 Introduction

Graph neural architecture search (GNAS), aiming to automatically discover the optimal architecture for graph neural network (GNN) based on graph-structured data and task, has shown remarkable progress in enhancing the predictive power and saving human endeavors for various graph applications [1]. The existing GNAS methods generally follow a supervised paradigm such that they optimize the weights within architectures given a training dataset with a supervised loss (e.g., the cross entropy loss of label predictions) and estimate the architecture performance based on the validation dataset with supervision signals. For example, the label prediction accuracy is adopted for architecture ranking during the neural architecture search process [2, 3, 4]. As a result, supervised labels become indispensable for applying the existing GNAS methods.

However, ground-truth labels in reality may be extremely scarce or hardly available in many graph applications. For example, a variety of biological problems require a significant amount of human labors and time costs in clinical tests to obtain labels for supervision [5, 6, 7]. As the existing

---

*This work was done during the author's internship at Wechat, Tencent
†Corresponding authors

37th Conference on Neural Information Processing Systems (NeurIPS 2023).

GNAS approaches heavily rely on supervised labels for weight training and architecture evaluation, they will suffer from performance deterioration in unsupervised settings, failing to discover optimal architectures in the scenarios where labels are scarce or not available.

In this paper, we study unsupervised graph neural architecture search, i.e., discovering optimal GNN architectures without labels for graph-structured data, which remains unexplored in the literature. The key problem lies in two important aspects: i) discover the latent graph factors that drive the formation process of graph data [8, 9, 10, 11, 12, 13, 14]; ii) capture the underlying relations between the factors and the optimal neural architectures. For instance, a molecular graph may consist of groups of atoms as well as bonds representing different functional units [15], requiring different optimal neural architectures to make accurate predictions.

Nevertheless, solving the problem is highly non-trivial and challenging given that the hidden factors are entangled in the graph and very difficult to capture, e.g., a social network may contain several communities originating from various interests (e.g., sports, games, etc.) [16, 10], with the nodes and edges belonging to different communities mixing together. Moreover, the architectures with different functional factors are also entangled within the weight-sharing super-network [17, 18, 19], resulting in inaccurate architecture performance estimations under different hidden factors.

To tackle the challenge, we propose a novel unsupervised graph neural architecture search method, i.e., Disentangled Self-supervised Graph Neural Architecture Search (**DSGAS**)[3]. Given graph data without supervised labels, our proposed **DSGAS** model can discover the optimal architectures capturing multiple latent factors in a self-supervised fashion. In particular, we first design a disentangled graph super-network, where multiple architectures are disentangled for simultaneous optimization w.r.t various latent factors. Then, we propose a self-supervised training with joint architecture-graph disentanglement, which disentangles architectures and graphs within a common latent space. The super-network is trained through a routing mechanism between architectures, graphs and self-supervised tasks, to obtain an accurate estimation of the architecture performance under each latent factor. Finally, we propose a contrastive search with architecture augmentations, where a novel architecture-level instance discrimination task is introduced to discover architectures with distinct capabilities of capturing various factors in a self-supervised fashion. Extensive experiments show that the proposed **DSGAS** model is able to significantly outperform the state-of-the-art GNAS baselines under both unsupervised and semi-supervised settings. Detailed ablation studies and analyses also demonstrate that **DSGAS** is able to discover effective architectures with our proposed disentangled self-supervision designs. The contributions of this paper are summarized as follows:

- We are the first to study the problem of unsupervised graph neural architecture search and propose the Disentangled Self-supervised Graph Neural Architecture Search (**DSGAS**) model capable of discovering the optimal architectures without labels, to the best of our knowledge.

- We introduce three novel modules, i) disentangled graph architecture super-network, ii) self-supervised training with joint architecture-graph disentanglement and iii) contrastive search with architecture augmentations, which can discover the optimal architectures capturing various graph latent factors with disentangled self-supervision.

- Extensive experiments on 11 real-world graph datasets show that our proposed method **DSGAS** is able to discover effective graph neural architectures without supervised labels and significantly outperform the state-of-the-art baselines in both unsupervised and semi-supervised settings.

## 2 Preliminaries and Problem Formulation

**Graph Neural Architecture Search** Denote the graph space as $\mathcal{G}$ and the label space as $\mathcal{Y}$. A graph neural network can be denoted as a function $f_{\alpha,w} : \mathcal{G} \rightarrow \mathcal{Y}$, which is characterized by architecture parameters $\alpha \in \mathcal{A}$ and learnable weights $w \in \mathcal{W}$ given an architecture space $\mathcal{A}$ and a weight space $\mathcal{W}$. Graph neural architecture search (GNAS) aims at automating the design of graph neural architectures, i.e., obtaining the best-performed architectures by searching $\alpha$. As $\alpha$ is usually instantiated as selecting GNN operations, e.g., GCN [20], GAT [21], GIN [22], we also call $\alpha$ as

---

[3]The codes are available at Github.

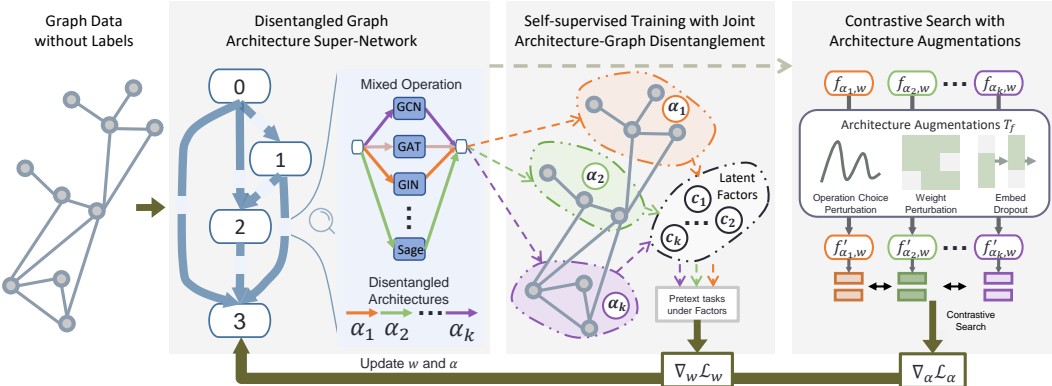

**Disentangled Self-Supervised Graph Neural Architecture Search**

Figure 1: The framework of Disentangled Self-supervised Graph Neural Architecture Search (**DSGAS**), including the following three key components: 1) Disentangled graph architecture super-network enables multiple architectures to be disentangled and optimized simultaneously in an end-to-end manner. 2) Self-supervised training with joint architecture-graph disentanglement estimates the performance of architectures under various latent factors by considering the relationship between architectures, graphs and factors. 3) Contrastive search with architecture augmentations encourages and discovers architectures with distinct capabilities of capturing factors. (Best viewed in color)

operation choices for brevity. Generally, GNAS solves the bi-level optimization problem [23] :

$$\alpha^* = \arg\min_{\alpha \in \mathcal{A}} \mathcal{L}_{\text{val}}(\alpha, w^*(\alpha)), \tag{1}$$

$$\text{s.t. } w^*(\alpha) = \arg\min_{w \in \mathcal{W}(\alpha)} \mathcal{L}_{\text{train}}(\alpha, w), \tag{2}$$

where $\mathcal{L}_{\text{train}}$ and $\mathcal{L}_{\text{val}}$ denotes the loss of the predictions of the architecture $f_{\alpha,w}(\cdot)$ against supervised labels on training and validation datasets. The optimization problem can be viewed as having two objectives that Eq.(2) aims to obtain accurate architecture performance estimation, and Eq.(1) aims to search the best-performed architectures. To avoid the cost of training from scratch for each architecture, the super-network [24, 25] arises as a commonly adopted technique to obtain faster architecture performance estimation, where the architecture candidates are viewed as sub-networks of the super-network, and their weights are shared during the training process.

**Unsupervised Graph Neural Architecture Search** We consider the problem of unsupervised graph neural architecture where labels, which are adopted for the performance estimation and the search process in the supervised GNAS, are not accessible. The problem of unsupervised GNAS can be formulated as optimizing an architecture generator that is able to discover powerful architectures by exploiting inherent graph properties without labels, i.e., $\mathcal{G} \mapsto f_{\alpha,w}$ instead of $(\mathcal{G}, \mathcal{Y}) \mapsto f_{\alpha,w}$ as done by supervised GNAS methods. Then, the discovered architectures $f_{\alpha,w}(\cdot)$ can be utilized in downstream tasks, e.g., finetuning the weights $w$ or the operation choices $\alpha$, or extra shallow classifiers for further prediction.

## 3 Disentangled Self-supervised Graph Neural Architecture Search

In this section, we introduce Disentangled Self-supervised Graph Neural Architecture Search (**DSGAS**) to search architectures without labels, by proposing three key components: disentangled graph architecture super-network, self-supervised training with joint architecture-graph disentanglement, and contrastive search with architecture augmentations.

### 3.1 Disentangled Graph Architecture Super-Network

To discover architectures that potentially have optimal performance, we resort to guiding the search towards architectures' capabilities of capturing the inherent graph factors, which are shown important in the graph formation [8, 9]. As architectures may expert in different graph factors, we propose a

disentangled graph architecture super-network to incorporate $K$ different architectures to be estimated and searched w.r.t factors simultaneously, where the hyperparameter $K$ denotes the number of factors.

**Disentangled Super-Network Layer**   For each super-network layer, we adopt $K$ mixed operations parameterized by different $\alpha$ to learn $K$-chunk graph representations:

$$\mathbf{H}_k \leftarrow \overline{\text{GNN}}_{\alpha_k}\left(\mathbf{H}, \mathbf{A}\right), \tag{3}$$

where $\mathbf{A}$ is the adjacency matrix of the graph, $\mathbf{H}$ denotes the input graph representations, and $\overline{\text{GNN}}_{\alpha_k}(\cdot)$ denotes the mixed GNN operations parameterized by $\alpha_k$. For the convenience of differentiable optimization, we adopt continuous parameterization and weight-sharing mechanism [25] to implement the mixed operations:

$$\overline{\text{GNN}}_{\alpha_k}(\mathbf{H}, \mathbf{A}) = \sum_{i=1}^{|\mathcal{O}|} \alpha_{k,i} \text{GNN}_i(\mathbf{H}, \mathbf{A}), \tag{4}$$

where $|\mathcal{O}|$ is the number of GNN operation choices, $\alpha_{k,i} = \frac{\exp(\theta_{\alpha_{k,i}})}{\sum_j \exp(\theta_{\alpha_{k,j}})}$ denotes the probability of the $i$-th operation for the $k$-th architecture $\alpha_k$, and $\theta$ is learnable parameters.

**Overall Disentangled Super-Network**   The overall super-network is constructed in the form of a directed acyclic graph (DAG) with an ordered sequence of disentangled super-network layers. More details about the DAG construction and the according GNN operations for each disentangled super-network layer are included in Appendix. The output of the last layer $\mathbf{Z} = [\mathbf{H}_1, \mathbf{H}_2, \ldots, \mathbf{H}_K]$ describes the various aspects of the graphs and serves as the final graph representations, which can be utilized or finetuned in downstream tasks. In this way, the architectures' operation choices $\alpha = [\alpha_1, \alpha_2, \ldots, \alpha_K]$ and weights $w$ are incorporated in one super-network. For brevity, we use $f_{\alpha_k,w}(\cdot)$ to denote the $k$-th architecture induced from the super-network. Note that the design of $K$ operation choices alleviates the entanglement of architectures by providing more flexible choices of paths [26] in the super-network. For instance, the 'mean' operation captures structural properties while the 'max' operation captures representative elements [22], and in this case, our design can capture both of them by choosing corresponding operations to learn respective representations instead of choosing only one of them which may conflict each other.

### 3.2   Self-supervised Training with Joint Architecture-Graph Disentanglement

Inspired by graph self-supervised learning, we utilize graph pretext tasks to measure the architectures' capabilities of capturing latent factors. Predictive pretext tasks [27], for example, design a pseudo label generator $s(\cdot)$, and optimize the prediction probability[4] $p(s(\mathcal{G}_i)|\mathcal{G}_i)$. However, the tasks usually take holistic views of the graphs and neglect the entanglement of the latent factors, which may lead to suboptimal performance estimation. Therefore, to disentangle the factors, we transform the probability into the expectation of multiple subtasks under different latent factors by the Bayesian formula:

$$p(s(\mathcal{G}_i)|\mathcal{G}_i) = \mathbb{E}_{p(k|\mathcal{G}_i)} p(s(\mathcal{G}_i)|\mathcal{G}_i, k), \tag{5}$$

where $p(k|\mathcal{G}_i)$ denotes the probability of latent factor $k$ given the $i$-th graph instance $\mathcal{G}_i$, and $p(s(\mathcal{G}_i)|\mathcal{G}_i, k)$ denotes the pretext task under $k$-th latent factor. An intuitive explanation of Eq. (5) is that it first infers the latent factors and then conducts factor-specific self-supervised training to capture the latent factors, which we describe in detail as follows.

**Architecture-aware Latent Factor Inference**   Directly modeling $p(k|\mathcal{G}_i)$ is difficult as we do not know prior what GNN encoders are suitable for inferring the latent factors. Intuitively, one solution is to get the architectures being searched involved in the inference stage. By the Bayesian formula, we factorize the probability w.r.t architecture choices:

$$p(k|\mathcal{G}_i) = \mathbb{E}_{p(\alpha_j|\mathcal{G}_i)} p(k|\mathcal{G}_i, \alpha_j), \tag{6}$$

---

[4]We take graph classification as an example for simplicity, while the case of node classification can be easily extended.

where $p(\alpha_j|\mathcal{G}_i)$ is a prior distribution, and we adopt a uniform distribution for simplicity. Then we can model the probability distributions of latent factors given the graph $\mathcal{G}_i$ by utilizing the architectures being searched:

$$p(k|\mathcal{G}_i, \alpha_j) = \frac{\exp \phi(\mathbf{z}_{i,j}||\text{Enc}(\alpha_j), \mathbf{c}_k)}{\sum_{m=1}^{K} \exp \phi(\mathbf{z}_{i,j}||\text{Enc}(\alpha_j), \mathbf{c}_m)}, \tag{7}$$

where $\mathbf{c}_k$ is a learnable vector to represent the $k$-th latent factor, and $\mathbf{z}_{i,k} = f_{\alpha_k,w}(\mathcal{G}_i)$ denotes the graph representations output by the $k$-th architecture for the graph $\mathcal{G}_i$. $\text{Enc}(\cdot)$ denotes architecture encoding techniques to obtain embeddings of operation choices $\alpha$ so that the structural properties and correlations of the neural architectures can be considered [28, 29].

**Factor-aware Graph Self-Supervised Learning**   Under the $k$-th latent factor, we leverage the corresponding architectures $f_{\alpha_k,w}(\cdot)$ for conducting the factor-specific pretext tasks as $p(s(\mathcal{G}_i)|\mathcal{G}_i, k)$ to estimate the capturing capabilities of architectures under various factors. The overall objective is to maximize Eq.(5), and the loss can be calculated by

$$\frac{1}{N} \sum_i -\log \mathbb{E}_{p(k|\mathcal{G}_i)}\Big(p(s(\mathcal{G}_i)|\mathcal{G}_i, k)\Big) \leq \frac{1}{N} \sum_i \mathbb{E}_{p(k|\mathcal{G}_i)}\Big(-\log p(s(\mathcal{G}_i)|\mathcal{G}_i, k)\Big), \tag{8}$$

where $N$ is the number of samples and the upper bound is obtained by Jensen's Inequality. Then we can generalize our method to other graph self-supervised tasks with specially-designed task loss functions by defining $-\log p(s(\mathcal{G}_i)|\mathcal{G}_i, k)$ as the task loss function $l(f_{\alpha_k,w}, \mathcal{G}_i)$ for the graph $\mathcal{G}_i$ under the $k$-th factor, and calculate the loss by

$$\mathcal{L}_w = \frac{1}{N} \sum_i \mathbb{E}_{p(k|\mathcal{G}_i)}\Big(l(f_{\alpha_k,w}, \mathcal{G}_i)\Big). \tag{9}$$

In this way, the disentangled architectures in Sec. 3.1 coupled with factors disentangled from graph data can be routed pairwisely, and trained with factor-specific self-supervision to obtain more accurate performance estimation under each factor. Similar to [25], the super-network weights are updated with $w = w - \lambda_w \nabla_w \mathcal{L}_w$ to obtain the weights that can represent the architectures' capabilities.

### 3.3   Contrastive Search with Architecture Augmentations

In this section, we focus on encouraging the disentanglement of architectures and searching architectures with distinct capabilities of capturing different factors. The main insight of our proposed search method is intuitively based on the following two observations shown in the literature: 1) As architectures similar in operation choices and topologies have similar capabilities of capturing semantics for downstream tasks [28, 30, 31], slight modifying the architecture will have a slight influence on its capability. 2) Since different GNN architectures expert in different downstream tasks [32], the architectures searched for different disentangled latent factors are expected to have dissimilar capabilities under different factors.

**Contrastive Search**   Inspired by self-supervised contrastive learning [33, 34] that capture discriminative features by pulling similar instances together and pushing dissimilar instances away in the latent space, we propose an architecture-level instance discrimination task to encourage the architectures to capture various latent factors. The task is defined as

$$p(s(\alpha_k)|\mathcal{G}_i, \alpha_k) = \frac{\exp \phi(\mathbf{z}_{i,k}, \mathbf{z}'_{i,s(\alpha_k)})}{\sum_{j=1}^{N} \exp \phi(\mathbf{z}_{i,j}, \mathbf{z}'_{i,s(\alpha_j)})}, \tag{10}$$

$$\mathbf{z}_{i,k} = f_{\alpha_k,w}(\mathcal{G}_i), \mathbf{z}'_{i,k} = T_f(f_{\alpha_k,w})(\mathcal{G}_i), \tag{11}$$

where $s(\alpha_k)$ is assigned to $k$ as surrogate labels for the architecture, $\phi(\cdot)$ calculates the similarity of two embeddings and $T_f(\cdot)$ denotes architecture augmentations that transform an architecture to another architecture with similar capabilities capturing factors, i.e., $f_{\alpha_k,w} \mapsto f'_{\alpha_k,w}$. Then the loss function can be calculated by

$$\mathcal{L}_\alpha = \sum_i -\log \mathbb{E}_{p(\alpha_k|\mathcal{G}_i)} p(s(\alpha_k)|\mathcal{G}_i, \alpha_k). \tag{12}$$

Similar to [25], the architecture parameters are updated with $\alpha = \alpha - \lambda_\alpha \nabla_\alpha \mathcal{L}_\alpha$ to search architectures with better capabilities of capturing factors.

**Architecture Augmentations**   To create various views of architectures, we design three basic architecture augmentations from the perspectives of architecture operation choices $\alpha$, architecture weights $w$ and the internal embeddings $\mathbf{H}$:

- Operation Choice Perturbation. This augmentation randomly reshapes the distributions of the mixed operations by altering the temperature in the softmax function in Eq. (4) with

$$\alpha_{k,i} = \frac{\exp(\theta_{\alpha_{k,i}}/\tau)}{\sum_j \exp(\theta_{\alpha_{k,j}}/\tau)}, \tag{13}$$

  where the temperature $\tau$ is sampled from a uniform distribution $\mathcal{U}([1/r_1, r_1])$, and $r_1 \geq 1$ is a hyper-parameter controlling the perturbation degree.
- Weight Perturbation. This augmentation randomly adds Gaussian noises $\epsilon \sim \mathcal{N}(0, \sigma^2)$ to $r_2\%$ of the architecture weights $w$, where $r_2$ controls the perturbation ratio, and $\sigma^2$ is the standard deviation of the weights.
- Embedding Dropout. This augmentation randomly drops $r_3\%$ of the embeddings $\mathbf{H}$ output from the mixed operations, where $r_3$ controls the dropout ratio.

Note that these architecture augmentations can further be randomly composed to generate mixed augmentations.

## 4   Experiments

In this section, we conduct experiments on 8 real-world datasets with unsupervised settings to verify the design of our method. We also include detailed ablation studies to analyze the effectiveness of each component, and 3 real-world datasets in semi-supervised settings to show that our method can alleviate the label scarcity issues by pretraining the super-network.

**Baselines**   We compare our method with 11 baselines from the following two different categories.

- **Manually designed GNNs.** We include five representative GNNs as our baselines, i.e., GCN [20], GAT [21], GIN [22], GraphSage [35], GraphConv [36] and a simple baseline MLP, which constitutes our search space. For graph-level classification tasks, we adopt global mean pooling for each layer and concatenation to obtain the graph representations for these baselines.
- **Graph neural architecture search.** We include representative GNAS baselines GraphNAS [3], PAS [4] and GASSO [37], where PAS is specially designed for graph-level classification tasks by searching the pooling operations, and GASSO is specially designed for node-level classification tasks by searching the graph structures simultaneously. We also include two classical NAS baselines, random search and DARTS [25]. As these baselines are not specially designed for graphs, we adopt our search space for these baselines.

Table 1: Summary of dataset statistics. Unsup./Semi. denotes Unsupervised and Semi-supervised settings. Graph/Node denotes graph and node classification tasks. ACC/AUC denotes Accuracy and ROC-AUC evaluation metrics.

| Datasets | MUTAG | IMDB-B | PROTEINS | DD | Computers | Photos | CS | Physics | OGBG-Molhiv | OGBN-Arxiv | Wechat-Video |
|---|---|---|---|---|---|---|---|---|---|---|---|
| Setting | Unsup. | Unsup. | Unsup. | Unsup. | Unsup. | Unsup. | Unsup. | Unsup. | Semi. | Semi. | Semi. |
| Task | Graph | Graph | Graph | Graph | Node | Node | Node | Node | Graph | Node | Node |
| Evaluation Metrirc | ACC | ACC | ACC | ACC | ACC | ACC | ACC | ACC | AUC | ACC | AUC |
| # Graphs | 188 | 1,000 | 1,113 | 1,178 | 1 | 1 | 1 | 1 | 41,127 | 1 | 1 |
| # Avg. Nodes | 17 | 19 | 39 | 284 | 13,752 | 7,650 | 18,333 | 34,493 | 26 | 169,343 | 60,774 |
| # Avg. Edges | 56 | 211 | 183 | 1,714 | 505,474 | 245,812 | 182,121 | 530,417 | 28 | 2,484,941 | 3,182,156 |
| # Features | 7 | 1 | 3 | 89 | 767 | 745 | 6,805 | 8,415 | 300 | 128 | 512 |
| # Classes | 2 | 2 | 2 | 2 | 10 | 8 | 15 | 5 | 2 | 40 | 13 |

**Datasets**   For unsupervised settings, we conduct experiments on four graph-level classification datasets including PROTEINS [38], DD [39], MUTAG [40], IMDB-B [41] from TUDataset [42] and four node-level classification datasets Coauthor CS, Coauthor Physics from the Microsoft Academic Graph [43], Amazon Computers, Amazon Photos from the Amazon Co-purchase Graph [44]. For semi-supervised settings, we adopt three real-world datasets, OGBG-Molhiv, OGBN-Arxiv [45]

Table 2: The results (accuracy%) of all the methods on the real-world datasets in unsupervised settings. Numbers after the ± signs represent standard deviations. The best results are in bold and the second-best results are underlined. As the search space of GASSO and PAS do not suit the graph-level and node-level tasks respectively, we omit their results.

| Data | Graph Classification | | | | Node Classification | | | |
| Method | PROTEINS | DD | MUTAG | IMDB-B | CS | Computers | Physics | Photo |
|---|---|---|---|---|---|---|---|---|
| GCN | 72.8±0.7 | 77.0±0.9 | 78.6±1.6 | 63.5±0.8 | 93.0±0.3 | 86.0±0.4 | 95.7±0.1 | 90.8±0.6 |
| GAT | 72.3±0.9 | 77.5±0.7 | 78.0±0.8 | 54.4±1.7 | 93.4±0.3 | 85.8±0.3 | 95.6±0.1 | 91.4±0.6 |
| GIN | 72.6±0.4 | 77.3±0.7 | 86.3±1.7 | 70.7±0.5 | 93.1±0.3 | 76.7±0.5 | 95.3±0.1 | 91.1±0.7 |
| GraphSage | 72.9±0.7 | 77.1±0.4 | 78.3±1.6 | 53.0±2.1 | 93.2±0.3 | 78.4±0.4 | 95.4±0.1 | 89.2±0.7 |
| GraphConv | 72.1±0.6 | 77.3±0.6 | 87.2±1.4 | 71.1±0.6 | 93.1±0.3 | 74.7±0.7 | 95.3±0.1 | 91.5±0.5 |
| MLP | 70.5±0.4 | 76.1±0.7 | 74.8±1.1 | 50.3±0.6 | 91.5±0.4 | 56.6±0.3 | 94.6±0.1 | 87.4±0.8 |
| Random | 74.5±0.9 | 74.8±1.3 | 82.1±2.8 | 69.0±2.1 | 92.9±0.3 | 84.8±0.4 | 95.4±0.1 | 91.1±0.6 |
| DARTS | 73.6±0.9 | 75.7±0.9 | 86.5±2.3 | 70.4±0.6 | 92.8±0.3 | 79.7±0.5 | 95.2±0.1 | 91.5±0.6 |
| GraphNAS | 73.6±0.7 | 75.2±0.9 | 77.5±0.7 | 62.7±1.3 | 91.6±0.3 | 69.0±0.6 | 94.5±0.1 | 89.3±0.7 |
| PAS | 74.6±0.3 | 76.5±0.9 | 84.0±1.6 | 64.6±13.8 | - | - | - | - |
| GASSO | - | - | - | - | 93.1±0.3 | 84.9±0.4 | 95.7±0.1 | 92.0±0.3 |
| DSGAS | **76.0±0.2** | **78.4±0.7** | **88.7±0.7** | **72.0±0.5** | **93.5±0.2** | **86.6±0.4** | **95.7±0.1** | **93.3±0.3** |

and Wechat-Video[5]. The datasets cover various graph-related fields including small molecules, bioinformatics, social networks, e-commerce networks, and academic coauthorship networks. The statistics are summarized in Table 1.

**Super-network**  We briefly introduce the super-network construction as follows. The super-network consists of two parts, the operation pool and the directed acyclic graph (DAG). The operation pool includes several node aggregation operations (e.g., GCN, GIN, GAT, etc.), graph pooling operations (e.g., SortPool, AttentionPool, etc.), and layer merging operations (e.g., MaxMerge, ConcatMerge, etc.). The DAG determines how the operations are connected to calculate the graph representations for the subsequent classification tasks.

More details of the experiments are provided in the Appendix, including additional experiments and analyses, experimental setups, configurations, and implementation details.

## 4.1 Main Results

**Unsupervised Settings**  From the experimental results summarized in Table 2, we have the following observations: 1) *GNNs perform differently on various datasets.* The best GNN baselines for these datasets are GraphSage, GAT, GraphConv, GraphConv, GAT, GCN, GCN, GraphConv successively, and the performance varies greatly across datasets and GNNs. It verifies that no GNN architecture is dominant for all datasets, which is consistent with the literature [32] and shows the demand for automated GNN architecture designing based on the data characteristics to obtain the optimal representations. 2) *Most GNAS baselines fail in the unsupervised setting.* Since the existing GNAS baselines highly rely on supervised signals to search the architectures, they inherently do not suit the unsupervised settings. As an ad hoc remedy for the existing GNAS baselines in unsupervised settings, we substitute the supervised metrics with the self-supervised ones during the searching process as simple extensions. However, these GNAS baselines, contrary to supervised settings, do not guarantee better performance than manually designed GNNs for all datasets. For example, all GNAS baselines even perform worse than manually designed GNNs on DD dataset and do not have significant improvements on most datasets. The reasons behind might be that simply using graph self-supervised metrics does not consider the entanglement of architectures and factors, leading to inaccurate estimation of the architectures' capabilities. 3) *Our method has significant improvements over the baselines on most datasets.* Compared with manually designed GNN baselines, **DSGAS** has a performance improvement of 3.1% on PROTEINS and over 1% on most datasets. We contribute this to its ability of automatically tailoring GNNs for various datasets, showing its effectiveness of automatic GNN designing. **DSGAS** also significantly surpasses GNAS baselines, showing its superiority in graph neural architecture search in unsupervised settings, which benefits from the design of discovering architectures that can capture various graph factors in a self-supervised fashion.

---

[5] https://algo.weixin.qq.com/

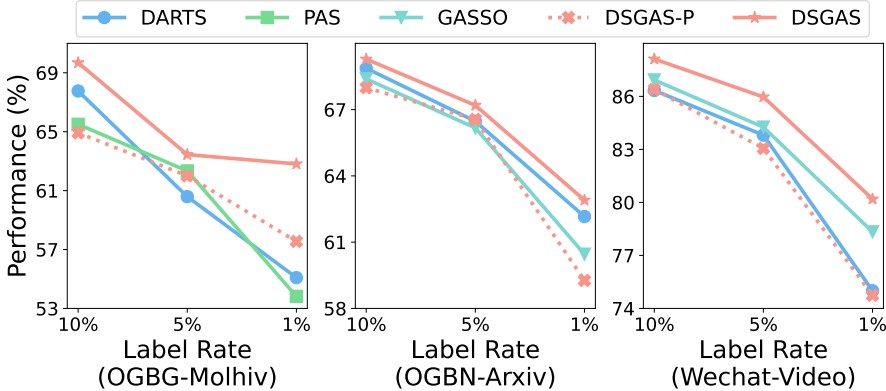

Figure 2: The performance of GNAS methods on real-world datasets under semi-supervised settings, where DSGAS-P denotes DSGAS without pretraining. The results are averaged by five random runs. (Best viewed in color)

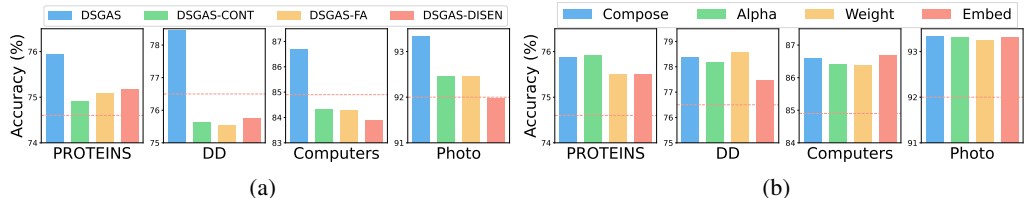

Figure 3: (a) Comparisons of different ablated variants of **DSGAS** on real-world datasets under unsupervised settings. The horizontal dashed line refers to the results of the best-performed GNAS baseline. (b) Comparisons of different architecture augmentations of **DSGAS** on real-world datasets under unsupervised settings, where 'Alpha', 'Weight' and 'Embed' denote the augmentations from perspective of operation choices, weight and embeddings. 'Compose' denotes uniformly choosing one of the three augmentations. The horizontal dashed line refers to the results of the best-performed GNAS baseline. (Best viewed in color)

**Semi-supervised Settings** From Figure 2, we have the following observations: Compared with the baselines, **DSGAS** significantly alleviates the performance drop when the number of available supervised labels is fewer, which verifies that our method fully exploits latent factors inside graph data and boost the supervised architecture search stage by warming up the weights and architecture parameters of the super-network. Its significant improvement over the ablated version DSGAS-P also verifies the effectiveness of pretraining the super-network by the proposed modules of self-supervised training with joint architecture-graph disentanglement and contrastive search with architecture augmentations. For example, our model with the pretraining stage has an absolute improvement of 5% on Wechat-Video dataset with 1% labels compared with the ablated version without the pretraining stage, which shows the effects of pretraining the super-networks on alleviating the label scarcity issues. In comparison, the performance of other baselines decays significantly more than our method, showing that current GNAS can not tackle scenarios with scarce labels. For example, on OGBG-Molhiv, PAS is the best baseline while the worst with 10% and 1% labels respectively, which may due to the inaccurate performance estimation with scarce labels. The phenomenon further strengthens the necessity of designing effective unsupervised GNAS methods.

## 4.2 Additional Experiments

**Ablation Studies** We evaluate the effectiveness of each module of our framework by comparing the following ablated versions of our method: DSGAS-CONT removes our proposed contrastive search module and search architectures with the vanilla self-supervised loss. DSGAS-FA further replaces our proposed factor-aware training module with the vanilla self-supervised training. DSGAS-DISEN further replaces our proposed disentangled super-network with the vanilla super-network.

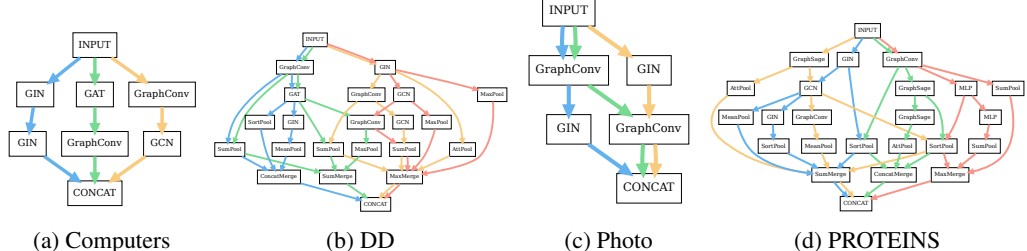

| (a) Computers | (b) DD | (c) Photo | (d) PROTEINS |

Figure 4: Visualizations of the search architectures on different datasets. Nodes denote GNN operations except that 'INPUT' denotes the input graphs with structures and features. Directed edges denote calculation flows, where different colors denote the architecture operation choices under different factors. (Best viewed in color)

We compare the performance of the ablated versions and the best GNAS baselines on the real-world datasets under unsupervised settings. From Figure 3a, we have the following observations. First, our proposed **DSGAS** outperforms all the variants on all datasets, demonstrating the effectiveness of each component of our proposed framework in searching graph architectures under unsupervised settings. Second, DSGAS-CONT drops drastically in performance on all datasets compared to the full version, showing the superiority of our proposed disentangled contrastive architecture search module in searching architectures. Third, the performance also decays for DSGAS-FA and DSGAS-DISEN on most datasets, showing the necessity of capturing various latent factors with different architectures.

**Architecture Visualizations**    We visualize the architectures searched on different unsupervised datasets in Figure 4. It shows that the searched architectures of different factors adopt quite different GNN operations while sometimes sharing the same operations, which leads to an overall architecture with complex internal connections between operations. This phenomenon implies that **DSGAS** can optimize the architecture operation choices as well as the operation connections for different factors to have a competitive performance on various graph datasets, which also verifies the superiority of **DSGAS** in automated architecture search to save human endeavors for architecture designs.

**Effects of Architecture Augmentations**    In Figure 3b, we show the results of different architecture augmentation methods on different datasets compared with the best GNAS baseline. We find that though the best augmentations differ among datasets, they have similar performance improvements in most cases, which verifies the design of contrastive search with architecture augmentations.

## 5   Related Works

**Graph Neural Architecture Search**    Instead of manually designing more sophisticated models for various scenarios, neural architecture search, aiming to automatically discover the optimal architectures for given tasks, emerges as a hot topic recently in computer vision [46, 23], natural language processing [47], graph representation learning [1, 48, 49], etc. In the field of graph representation learning with various applications [50, 51, 52, 53, 54], graph neural architecture search (GNAS) methods, as the most related to our works, can be roughly classified into reinforcement-learning-based methods [3, 55], evolutionary-based methods [56, 2, 57, 58], and differentiable methods [59, 60, 61, 62, 37, 63, 4, 64, 65]. However, supervised labels are indispensable for the existing GNAS methods to conduct neural architecture search, which limits their applications in widely-existed scenarios where labels are scarce or not available.

**Unsupervised Neural Architecture Search**    In unsupervised settings, some neural architecture search methods replace supervised labels with self-supervised loss during searching [66, 67, 68, 69, 70, 71]. Another classic of related methods design special metrics, whose calculation does not depend on labels, as proxies for model performance estimation [72, 73]. For example, UnNAS [74] adopts pretext tasks like image rotation, coloring images, solving puzzles, etc. However, these methods are specially designed for computer vision, and can not be directly adopted to graph data. Some GNAS

works exploit self-supervised loss as auxiliaries to augment the supervised search process [63], but the supervised labels are still mandatory for its search. To the best of our knowledge, this is the first work on unsupervised graph neural architecture search.

**Graph Self-supervised Learning**    Graph self-supervised learning [75, 76] is devoted to obtaining graph representations by extracting informative knowledge with well-designed pretext tasks without labels, which can be roughly classified into contrastive [77, 78, 79, 80, 81, 82, 83, 84, 85, 86] and generative [87, 88, 76, 89], and predictive methods [90, 91, 92]. The existing graph self-supervised learning methods usually focus on designing better pretext tasks with a fixed encoder such as GCN [20], GAT [21] and GIN [22]. Another class of related methods attempt to automate the choices of pretext tasks [93, 94, 95, 96, 97]. We mainly consider graph neural architecture in unsupervised settings, while other pretext tasks are orthogonal to our framework and can be incorporated.

**Disentangled Representation Learning**    The primary objective of disentangled representation learning is to delineate and interpret the various latent factors which influence the data we encounter in an observable context, rendering each of these factors as unique vector representations [98, 99]. It has emerged to be a useful tool in various domains, including those of computer vision [100, 101, 102, 103, 104, 105], and recommendation systems [106, 107, 108, 109, 110, 111, 112, 113], graph representation learning [9, 114, 10, 14, 11, 115, 116, 117]. As the most related, GRACES [63] characterize the graph latent factors inside data by designing a self-supervised disentangled graph encoder, and conduct graph neural architecture search for each graph to handle graph distribution shifts, while the training and searching process still follows the supervised paradigm. In contrast, we focus on automating the GNN designs with disentangled self-supervision in this paper.

## 6  Conclusions

In this paper, we propose a novel Disentangled Self-Supervised Graph Neural Architecture Search (**DSGAS**) framework to automate the GNN designs with disentangled self-supervision, which includes disentangled graph architecture super-network, self-supervised training with joint architecture-graph disentanglement and contrastive search with architecture augmentations. Extensive experiments demonstrate that our proposed method can discover architectures with capabilities of capturing various graph latent factors and significantly outperform the state-of-the-art GNAS baselines. Detailed ablation studies and analyses show the effectiveness of our method design. One limitation is that in this paper we mainly focus on homogeneous graphs, and we leave extending our method to heterogeneous graphs for further explorations.

## Acknowledgements

This work was supported by the National Key Research and Development Program of China No. 2020AAA0106300, National Natural Science Foundation of China (No. 62222209, 62250008, 62102222, 62206149), Beijing National Research Center for Information Science and Technology under Grant No. BNR2023RC01003, BNR2023TD03006, China National Postdoctoral Program for Innovative Talents No. BX20220185, China Postdoctoral Science Foundation No. 2022M711813, and Beijing Key Lab of Networked Multimedia. All opinions, findings, conclusions, and recommendations in this paper are those of the authors and do not necessarily reflect the views of the funding agencies.

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
