# Unsupervised Graph Neural Architecture Search with Disentangled Self-supervision
## (Appendix)

**Zeyang Zhang**[1][*], **Xin Wang**[1][†], **Ziwei Zhang**[1], **Guangyao Shen**[2], **Shiqi Shen**[2], **Wenwu Zhu**[1][†]

[1]Department of Computer Science and Technology, BNRist, Tsinghua University, [2]Wechat, Tencent
zy-zhang20@mails.tsinghua.edu.cn, {xin_wang, zwzhang}@tsinghua.edu.cn,
thusgy2012@gmail.com, shiqishen@tencent.com, wwzhu@tsinghua.edu.cn

## A  Notations

Table 1: Summary of notations and their descriptions.

| Notations | Descriptions |
|---|---|
| $\mathcal{G}, \mathcal{Y}, \mathcal{A}, \mathcal{W}$ | Graph space, label space, architecture space and weight space |
| $\alpha, w$ | Architecture operation choices and architecture weight |
| $\mathbf{H}, \mathbf{A}$ | Graph embeddings and adjacency matrix |
| $\mathcal{O}, \|\mathcal{O}\|$ | A pool of GNN operations and the number of operations |
| $f_{\alpha,w}(\cdot)$ | A GNN characterized by operation choices $\alpha$ and weight $w$ |
| $s(\cdot)$ | Pseudo label generator defined by pretext tasks |
| $\mathbf{c}$ | Learnable vectors for the latent factors |
| $K$ | The number of the latent factors |
| $r$ | Perturbation ratio in architecture augmentations |
| $T_f(\cdot)$ | Architecture augmentation function |
| $\text{Enc}(\cdot)$ | Architecture encoding function |
| $l, \mathcal{L}$ | loss functions |
| $\phi(\cdot)$ | A function that calculates the similarity of two embeddings |

## B  Additional Experiments and Analyses

### B.1  Complexity Analysis

Denote the number of nodes and edges in the graph as $N$ and $E$, the number of latent factors as $K$, the number of operation choices as $|\mathcal{O}|$, the dimensionality of hidden representations as $d$. The time complexity of the disentangled super-network is $O(K|E|d + K|V|d^2)$, where the computation for each factor is fully parallelizable and amenable to GPU acceleration, and $K$ is usually a small constant. The time complexity of the self-supervised training and contrastive search modules is both $O(K^2 d^2)$. As architectures under different factors share the parameters, the number of learnable parameters is the same as classical graph super-network, i.e., $O(|\mathcal{O}|d^2)$. Therefore, the complexity of our method is comparable to classical GNAS methods.

37th Conference on Neural Information Processing Systems (NeurIPS 2023).

Table 2: Comparisons of NAS methods in terms of empirical running time and performance on unsupervised graph classification datasets (with single NVIDIA GeForce RTX 3090).

| Data | PROTEINS | | DD | | MUTAG | | IMDB-B | |
|---|---|---|---|---|---|---|---|---|
| Metric | ACC(%) | Time(s) | ACC(%) | Time(s) | ACC(%) | Time(s) | ACC(%) | Time(s) |
| Random | $74.5_{\pm0.9}$ | 2952 | $74.8_{\pm1.3}$ | 9401 | $82.1_{\pm2.8}$ | 949 | $69.0_{\pm2.1}$ | 2535 |
| DARTS | $73.6_{\pm0.9}$ | 80 | $75.7_{\pm0.9}$ | 650 | $86.5_{\pm2.3}$ | 21 | $70.4_{\pm0.6}$ | 65 |
| GraphNAS | $73.6_{\pm0.7}$ | 1897 | $75.2_{\pm0.9}$ | 7830 | $77.5_{\pm0.7}$ | 273 | $62.7_{\pm1.3}$ | 1595 |
| PAS | $\underline{74.6_{\pm0.3}}$ | 156 | $76.5_{\pm0.9}$ | 931 | $84.0_{\pm1.6}$ | 36 | $64.6_{\pm13.8}$ | 127 |
| GASSO | - | | - | | - | | - | |
| **DSGAS** | $\mathbf{76.0_{\pm0.2}}$ | 471 | $\mathbf{78.4_{\pm0.7}}$ | 1800 | $\mathbf{88.7_{\pm0.7}}$ | 41 | $\mathbf{72.0_{\pm0.5}}$ | 261 |

Table 3: Comparisons of NAS methods in terms of empirical running time and performance on unsupervised node classification datasets (with single NVIDIA GeForce RTX 3090).

| Data | CS | | Computers | | Physics | | Photo | |
|---|---|---|---|---|---|---|---|---|
| Metric | ACC(%) | Time(s) | ACC(%) | Time(s) | ACC(%) | Time(s) | ACC(%) | Time(s) |
| Random | $92.9_{\pm0.3}$ | 1071 | $84.8_{\pm0.4}$ | 3605 | $95.4_{\pm0.1}$ | 2095 | $91.1_{\pm0.6}$ | 522 |
| DARTS | $92.8_{\pm0.3}$ | 34 | $79.7_{\pm0.5}$ | 79 | $95.2_{\pm0.1}$ | 75 | $91.5_{\pm0.6}$ | 13 |
| GraphNAS | $91.6_{\pm0.3}$ | 647 | $69.0_{\pm0.6}$ | 5295 | $94.5_{\pm0.1}$ | 2268 | $89.3_{\pm0.7}$ | 435 |
| PAS | - | | - | | - | | - | |
| GASSO | $93.1_{\pm0.3}$ | 34 | $84.9_{\pm0.4}$ | 69 | $\underline{95.7_{\pm0.1}}$ | 75 | $\underline{92.0_{\pm0.3}}$ | 13 |
| **DSGAS** | $\mathbf{93.5_{\pm0.2}}$ | 49 | $\mathbf{86.6_{\pm0.4}}$ | 201 | $\mathbf{95.7_{\pm0.1}}$ | 99 | $\mathbf{93.3_{\pm0.3}}$ | 20 |

## B.2 Empirical Running Time

We make the comparisons of different NAS methods in terms of the empirical running time. The time is tested with one NVIDIA 3090 GPU. As shown in the Table 2 and Table 3, the running time of our method **DSGAS** is on par with the state-of-the-art one-shot NAS methods (e.g., DARTS, GASSO, and PAS), which is much more efficient than the multi-trial NAS methods (e.g., random search and GraphNAS). While being competitive in efficiency, our method has significant performance improvements over the baselines. The empirical results also confirm the theoretical complexity analysis in Section B.1 that our method does not introduce many additional computational costs.

## B.3 Search Space Analysis

We show how the performance of **DSGAS** changes when the search space is larger on the Computers dataset in the Table 4 and Table 5. In Table 4 , we enlarge the search space by gradually increasing the GNN operation pool , i.e. increasing the number of available GNN options. In Table 5, we enlarge the search space by gradually increasing the number of factors , i.e. increasing the number of paths to capture graph factors. As shown in the tables, when the search space is larger, the performance of our method gradually improves, which verifies that our method can discover better architectures with a larger search space.

## B.4 Discussions of the Searched Architectures

We visualize several searched architectures in Figure 4 of the main paper, which are powerful yet complex. Here, we make following discussions about the searched architectures.

---

*This work was done during the author's internship at Wechat, Tencent

†Corresponding authors

Table 4: The performance of our method with increasing number of available GNN options on the Computers dataset.

| $|\mathcal{O}|$ | 2 | 3 | 4 | 5 |
|---|---|---|---|---|
| ACC(%) | $84.8_{\pm0.4}$ | $86.2_{\pm0.3}$ | $86.6_{\pm0.3}$ | $86.6_{\pm0.4}$ |

Table 5: The performance of our method with increasing number of available factors on the Computers dataset.

| $K$ | 2 | 3 | 4 | 5 |
|---|---|---|---|---|
| ACC(%) | $85.1_{\pm 0.4}$ | $86.6_{\pm 0.4}$ | $87.3_{\pm 0.4}$ | $86.5_{\pm 0.4}$ |

Table 6: The performance of NAS methods on other graph datasets in unsupervised settings.

| Data | Cora | Citeseer | Pubmed |
|---|---|---|---|
| DARTS | $78.4_{\pm 0.3}$ | $71.1_{\pm 0.8}$ | $78.8_{\pm 0.7}$ |
| GraphNAS | $81.5_{\pm 0.6}$ | $70.4_{\pm 1.1}$ | $79.9_{\pm 0.9}$ |
| GASSO | $80.2_{\pm 0.8}$ | $69.5_{\pm 1.1}$ | $78.1_{\pm 0.8}$ |
| **DSGAS** | $\mathbf{83.5_{\pm 0.4}}$ | $\mathbf{72.2_{\pm 1.4}}$ | $\mathbf{80.6_{\pm 0.4}}$ |

We observe that in different random training runs, while the searched architectures show similar performance, their DAGs are not the same, which is consistent with the NAS literature [1, 2]. A possible reason is that there exist plenty of different architectures with very similar performance in the large graph architecture search space [3]. As shown in Table 2 in the main paper, our method has relatively low performance variance and high performance expectation, which shows that our method can better search for the potential top-ranked architectures than baselines.

The number of factors $K$, which reflects the assumption of the number of graph factors to be captured inside the data, controls the searchable architectures in our method. When $K = 1$, our method can include single simple architectures with arbitrary operation combinations. When $K \geq 1$, our method can discover more sophisticated architectures to capture the inherent graph factors and obtain better performance. Empirically, we observe that when $K \geq 1$, the searched architectures are more complex than single architectures but are also more competitive in capturing the graph properties, which verifies the design of our method.

### B.5 Additional Results in Unsupervised Settings

We provide the experimental results on Cora, CiteSeer, and PubMed in Table 6. We follow the public data splits [4] of Cora, Citeseer, and Pubmed, and conduct graph neural architecture search without labels. Similar to other unsupervised node classification datasets in the paper, we train the super-network with fixed epochs , and for evaluation, we train a linear classifier and report the mean accuracy and standard deviations on the test nodes of 5 runs with different random seeds. As shown in the Table 6, our method **DSGAS** has significant performance improvement over the NAS baselines.

## C  Experimental Details

### C.1  Unsupervised Settings

**Setups**  Following previous works of graph self-supervised learning [5, 6], we first pretrain the models by self-supervised loss with fixed epochs, and then evaluate the models by finetuning an extra classifier. As supervised labels are not available in unsupervised settings, for fair comparisons, all the methods adopt the same self-supervised tasks, [5] and [6] for graph and node classification tasks respectively. For GNAS baselines, the self-supervised loss is utilized to train the model parameters as well as select the architectures.

**Evaluation protocols**  For graph-level classification tasks, the obtained graph representations are evaluated by an SVM classifier with a 10-fold cross-validation and the process is repeated by five times with different seeds. For node-level classification tasks, the obtained node representations are evaluated by a logistic regression classifier with random splits twenty times. The average accuracies and their standard deviations are reported. These protocols are kept the same for all methods to guarantee fair comparisons. We summarize the pipeline of unsupervised settings for **DSGAS** in Algorithm 1.

---

**Algorithm 1** The pipeline of unsupervised settings for **DSGAS**

---

**Require:** Graph $\mathcal{G}$ without labels, training epochs $L$.

1: Construct the dientangled graph architecture super-networks with randomly initialized weights $w$ and operation choices $\alpha$.
2: **for** $l = 1, \ldots, L$ **do**
3:    Calculate the self-supervised training loss with architecture-graph disentanglement $\mathcal{L}_w$ as Eq. (9)
4:    Update the super-network weights with $w = w - \lambda_w \nabla_w \mathcal{L}_w$
5:    Calculate the contrastive search loss with architecture augmentations $\mathcal{L}_\alpha$ as Eq. (12)
6:    Update the super-network operation choices with $\alpha = \alpha - \lambda_\alpha \nabla_\alpha \mathcal{L}_\alpha$
7: **end for**
8: Evaluate the searched model with linear protocols.

---

## C.2   Semi-supervised Settings

**Setups**   To further test the performance of GNAS in scenarios with scarce labels instead of exactly no labels, we conduct semi-supervised experiments with limited labels, i.e., using 10%,5%,1% labels for both training and validation. In this setting, we compare the differentiable GNAS baselines, including DARTS, PAS and GASSO, which train super-network weights with training datasets and optimize architecture parameters with validation datasets as in supervised settings. For **DSGAS**, it first pretrains the super-network by methods mentioned in Section 3.2 and Section 3.3, and then continues the search process as traditional supervised GNAS. We also include DSGAS-P, which does not adopt the pretraining stage, as an ablated baseline.

**Evaluation protocols**   For OGBG-Molhiv and OGBN-Arxiv, the splits are the same in the open graph benchmark [7]. For Wechat-Video, we adopt random splits with a ratio of 6:2:2 for training, validation, and testing by multi-label stratified splitting [8]. The available training and validation labels are randomly sampled with a stratified sampling for settings of labeling rates 1%, 5%, and 10%. We train the models with early-stop patience 50, and then we adopt the best-performed checkpoint on validation data split, which is tested on testing data split to obtain the reported results. These splits and the training strategies are kept the same for all methods to guarantee fair comparisons. The experiments are run five times with different random seeds. We summarize the pipeline of semi-supervised settings for **DSGAS** in Algorithm 2.

---

**Algorithm 2** The pipeline of semi-supervised settings for **DSGAS**

---

**Require:** Graph $\mathcal{G}$ with limited labels, training epochs $L$, earlystop patience $E$.

1: Construct the dientangled graph architecture super-networks with randomly initialized weights $w$ and operation choices $\alpha$.
2: Pretraining the super-networks for $w$ and $\alpha$ using Algorithm 1
3: **for** $l = 1, \ldots, L$ **do**
4:    Calculate the supervised loss on training data $\mathcal{L}_w$.
5:    Update the super-network weights with $w = w - \lambda_w \nabla_w \mathcal{L}_w$.
6:    Calculate the supervised loss on validation data $\mathcal{L}_\alpha$.
7:    Update the super-network operation choices with $\alpha = \alpha - \lambda_\alpha \nabla_\alpha \mathcal{L}_\alpha$.
8:    **if** the validation accuracy is non-increasing for $E$ epochs **then**
9:       break
10:    **end if**
11: **end for**
12: Evaluate the searched model on testing data.

---

# D  Implementation Details

## D.1  Super-network Construction

The super-network generally consists of two parts, the operation pool and the directed acyclic graph (DAG) that wires the operations. Following [9], we adopt three kinds of operations in the operation pool as follows:

- Node aggregation operations, which aggregate messages from the neighborhood to update the node representations, including GCN [10], GAT [11], GIN [12], GraphConv [13], GraphSage [14]. MLP (Multi-layer Perceptrons) is also included as an operation that does not utilize the neighborhood.
- Graph pooling operations, which aggregate node representations to obtain graph-level representations, including SortPool [15], AttentionPool [16], MaxPool, MeanPool and SumPool. For example, MeanPool takes the average of the node representations as the graph representation.
- Layer merging operations, which aggregate representations from intermediate layers to formulate more expressive representations, including MaxMerge, ConcatMerge, SumMerge and MeanMerge. For example, MaxMerge selects the max values in multiple representations from intermediate layers.

For brevity, we denote 'Agg', 'Pool', 'Merge' as node aggregation operations, graph pooling operations, and layer merging operations respectively. Following [17], the DAG for node classification tasks is a straightforward path, i.e., $\mathbf{H}^{l+1} = \text{Agg}^l(\mathbf{H}^l, \mathbf{A})$, and the embeddings of the last layer are utilized for downstream tasks, where $\mathbf{H}^l$ denotes the hidden embeddings output by the $l$-th layer, and $\mathbf{A}$ denotes the graph adjacency matrix. Following [9], the DAG for graph classification tasks is constructed by $\mathbf{H}^{l+1} = \text{Agg}^l(\mathbf{H}^l, \mathbf{A}), \mathbf{Z}^l = \text{Pool}^l(\mathbf{H}^l, \mathbf{A})$, and the merged representations $\text{Merge}(\mathbf{Z}^1, \mathbf{Z}^2, \ldots, \mathbf{Z}^L)$ are utilized for downstream tasks, where $L$ is the number of layers.

## D.2  Hyperparameters

For fair comparisons, all methods adopt the same dimensionality, number of layers and normalization techniques. For graph classification datasets, we adopt the dimensionality as 32, the number of layers as 3 and batch normalization [18]. For node classification datasets, we adopt the dimensionality as 128 and the number of layers as 2 and layer normalization [19]. Adam optimizer [20] is adopted to optimize the model weights and another SGD optimizer is adopted to optimize architecture parameters for NAS methods. For our method, we adopt $K = 3$ for all node classification datasets and $K = 4$ for all graph classification datasets, and the hyperparameters that control the perturbation degree for the architecture augmentations of operation choices, weights and embeddings are set to 1.1, 0.1, 0.05 respectively for all datasets.

## D.3  Configurations

All experiments are conducted with:

- Operating System: Ubuntu 20.04.5 LTS
- CPU: Intel(R) Xeon(R) Gold 6240 CPU @ 2.60GHz
- GPU: NVIDIA GeForce RTX 3090 with 24 GB of memory
- Software: Python 3.9.12, Cuda 11.3, PyTorch [21] 1.12.1, PyTorch Geometric [22] 2.0.4.