# OpenReview forum: "Unsupervised Graph Neural Architecture Search with Disentangled Self-Supervision"
_NeurIPS.cc/2023/Conference — NeurIPS 2023 poster_

### Official Review · Reviewer_JBpb · 2023-07-06

**Soundness:** 2 fair
**Presentation:** 3 good
**Contribution:** 2 fair
**Rating:** 4
**Confidence:** 5

**Summary:**

This paper proposed a method DSGAS to automcatiicaly design the architectures in an unsupervised manner. It discovers the optimal  architectures by learning the latent graph factor.

**Strengths:**

It is interesting and novel to automate the design of GNNs in an unsupervised manner, which holds significant potential.


**Weaknesses:**

1. The necessity of designing disentanglement is not clear. It is true that GNNs may preferred different graph factors. What is the difference between the proposed method and the baseline that: using one latent factor design one architectures, and then run multiple times and ensembles the search architectures?
The motivation for disentangled architectures is not well justified. While the evaluations on K are provided, it would be more effective to achieve K=2 by ensemble two K=1 results.


2. The experiments are insufficient to justify the effectiveness of the proposed design method. Since it designs GNNs in an unsupervised manner, it would be better to conduct fair comparisons with existing unsupervised methods rather than methods designed under a supervised manner.

3. Apart from the comparisons of performance, the comparisons of search cost, the parameters of the searched architectures should be provided as well.

**Questions:**

Please check the weaknesses.

**Limitations:**

Please check the weaknesses.

---

> ### Author Rebuttal · Authors · 2023-08-09
>
> We sincerely thank the reviewer for the comments and suggestions. We have carefully reviewed each point raised and make responses to the reviewer point by point as follows.
>
> > The necessity of designing disentanglement is not clear. It is true that GNNs may preferred different graph factors. What is the difference between the proposed method and the baseline that: using one latent factor design one architectures, and then run multiple times and ensembles the search architectures? The motivation for disentangled architectures is not well justified. While the evaluations on K are provided, it would be more effective to achieve K=2 by ensemble two K=1 results.
>
> Thank you for your comment. Following your suggestion, we compare a baseline that ensembles K searched architectures by averaging their outputs. The results on the dataset Computers are shown in the following table.
>
> | K        | 2          | 3          | 4          | 5          |
> | :------- | :--------- | :--------- | :--------- | :--------- |
> | Ours     | 85\.1+-0.4 | 86\.7+-0.4 | 87\.3+-0.4 | 86\.5+-0.4 |
> | Ensemble | 84\.0+-0.3 | 84\.2+-0.4 | 84\.0+-0.5 | 83\.2+-0.4 |
>
> As shown in the table, our method has a significant performance improvement over the ensemble baseline. The reason might be that the ensemble baseline can not learn to capture various graph factors, and accordingly can not search the optimal architectures with regard to different graph factors. In contrast, our method is an end-to-end manner, which is learned to jointly discover the latent factors and search multiple factor-wise expert architectures to achieve state-of-the-art performance on various datasets.
>
> > It would be better to conduct comparisons with existing unsupervised methods rather than methods designed under a supervised manner.
>
> Thank you for your comment. To the best of our knowledge, the study of unsupervised graph neural architecture search remains unexplored in the literature, and our method is the first GNAS method designed under unsupervised settings. The existing GNAS methods are designed in a supervised manner, and for fair comparisons, we extend these methods as baselines by replacing the supervised loss with the self-supervised loss to suit the unsupervised NAS settings.
>
> > Apart from the comparisons of performance, the comparisons of search cost, the parameters of the searched architectures should be provided as well.
>
> Thank you for your suggestion. Following your suggestion, we provide the comparisons of GNAS methods in terms of performance, search cost, and architecture parameters in the following table. The time is tested with one NVIDIA 3090 GPU.
>
> | Data | CS |  |  | Computers |  |  | Physics |  |  | Photo |  |  |
> |:---:|:---:|:---:|:---:|:---:|:---:|:---:|:---:|:---:|:---:|:---:|:---:|:---:|
> | Metric | ACC(%) | Time(s) | #Params(K) | ACC(%) | Time(s) | #Params(K) | ACC(%) | Time(s) | #Params(K) | ACC(%) | Time(s) | #Params(K) |
> | Random | 92.9+-0.3 | 1071 | 899 | 84.8+-0.4 | 3605 | 144 | 95.4+-0.1 | 2095 | 1096 | 91.1+-0.6 | 522 | 142 |
> | DARTS | 92.8+-0.3 | 34 | 915 | 79.7+-0.5 | 79 | 144 | 95.2+-0.1 | 75 | 1116 | 91.5+-0.6 | 13 | 126 |
> | GraphNAS | 91.6+-0.3 | 647 | 1011 | 69.0+-0.6 | 5295 | 372 | 94.5+-0.1 | 2268 | 1198 | 89.3+-0.7 | 435 | 238 |
> | GASSO | 93.1+-0.3 | 34 | 2632 | 84.9+-0.4 | 69 | 370 | 95.7+-0.1 | 75 | 3236 | 92.0+-0.3 | 13 | 361 |
> | Ours | 93.5+-0.2 | 49 | 1013 | 86.6+-0.4 | 201 | 259 | 95.7+-0.1 | 99 | 1250 | 93.3+-0.3 | 20 | 288 |
>
>  As shown in the table above, the training time of our method is on par with the state-of-the-art one-shot NAS methods (e.g.,  DARTS, GASSO,), which is much more efficient than the multi-trial NAS methods (e.g., random and GraphNAS). The numbers of parameters of the searched architectures are comparable for different methods. While being competitive in efficiency, our method has significant performance improvements over the baselines.

---

> > ### Comment · Reviewer_JBpb · 2023-08-17
> >
> > Your response addressed most of my concerns.
> > For W2, the comparisons with unsupervised(or self-supervised) human-designed GNN methods are expected, rather than the unsupervised Graph NAS baselines.

---

> > > ### Author Response · Authors · 2023-08-17
> > >
> > > We thank the reviewer for the feedback. Following your suggestion, we provide the comparisons with representative self-supervised human-designed GNN methods. Specifically, for graph classification, we compare with DGK[1], Graph2Vec[2], InfoGraph[3], GraphCL[4], and JOAO[5] on PROTEINS dataset. For node classification, we compare with DGI[6], MVGRL[7], GRACE[8], GCA[9], and LaGraph[10] on CS dataset. The results are shown in the following tables.
> > >
> > > Table 1.
> > > | DGK[1]       | Graph2Vec[2] | InfoGraph[3] | GraphCL[4]   | JOAO[5]      | Ours      |
> > > |-----------|-----------|-----------|-----------|-----------|-----------|
> > > | 73.3+-0.8 | 73.3+-2.0 | 74.4+-0.3 | 74.4+-0.5 | 74.6+-0.4 | 76.0+-0.2 |
> > >
> > > Table 2.
> > > | DGI[6]       | MVGRL[7]     | GRACE[8]     | GCA[9]       | LaGraph[10]   | Ours      |
> > > |-----------|-----------|-----------|-----------|-----------|-----------|
> > > | 92.2+-0.6 | 92.1+-0.1 | 92.9+-0.0 | 93.1+-0.0 | 93.3+-0.2 | 93.5+-0.2 |
> > >
> > > The results show that our method has significant performance improvements over the self-supervised human-designed GNN baselines. We will incorporate the results in the revision.
> > >
> > > Once again, we thank you for your time and consideration. Should you have any further questions, we would be delighted to provide further responses.
> > >
> > >
> > > [1] Yanardag, Pinar, and S. V. N. Vishwanathan. "Deep graph kernels." Proceedings of the 21th ACM SIGKDD international conference on knowledge discovery and data mining. 2015.
> > >
> > > [2] Narayanan, Annamalai, et al. "graph2vec: Learning distributed representations of graphs." arXiv preprint arXiv:1707.05005 (2017).
> > >
> > > [3] Sun, Fan-Yun, et al. "InfoGraph: Unsupervised and Semi-supervised Graph-Level Representation Learning via Mutual Information Maximization." International Conference on Learning Representations. 2019.
> > >
> > > [4] You, Yuning, et al. "Graph contrastive learning with augmentations." Advances in neural information processing systems 33 (2020): 5812-5823.
> > >
> > > [5] You, Yuning, et al. "Graph contrastive learning automated." International Conference on Machine Learning. PMLR, 2021.
> > >
> > > [6] Veličković, Petar, et al. "Deep Graph Infomax." International Conference on Learning Representations. 2018.
> > >
> > > [7] Hassani, Kaveh, and Amir Hosein Khasahmadi. "Contrastive multi-view representation learning on graphs." International conference on machine learning. PMLR, 2020.
> > >
> > > [8] Zhu, Yanqiao, et al. "Deep graph contrastive representation learning." arXiv preprint arXiv:2006.04131 (2020).
> > >
> > > [9] Zhu, Yanqiao, et al. "Graph contrastive learning with adaptive augmentation." Proceedings of the Web Conference 2021. 2021.
> > >
> > > [10] Xie, Yaochen, Zhao Xu, and Shuiwang Ji. "Self-supervised representation learning via latent graph prediction." International Conference on Machine Learning. PMLR, 2022.

---

### Official Review · Reviewer_GehM · 2023-07-07

**Soundness:** 3 good
**Presentation:** 2 fair
**Contribution:** 3 good
**Rating:** 8
**Confidence:** 4

**Summary:**

This paper addresses the problem of unsupervised graph neural architecture search, which has received limited attention in existing literature. The authors propose a novel approach called Disentangled Self-supervised Graph Neural Architecture Search (DSGAS) to discover optimal architectures capturing latent graph factors in an unsupervised manner. The DSGAS model consists of a disentangled graph super-network that incorporates multiple architectures with factor-wise disentanglement, self-supervised training to estimate architecture performance under different factors, and a contrastive search method to discover architectures with factor-specific expertise. Extensive experiments on 11 real-world datasets demonstrate that the proposed model achieves state-of-the-art performance compared to several baseline methods.


**Strengths:**

- The paper addresses an important and underexplored problem in the field of graph neural architecture search, i.e., scenarios where supervised labels are not available.

- The proposed DSGAS model introduces a novel approach to discovering optimal architectures by leveraging latent graph factors in a self-supervised fashion based on unlabeled graph data.

- The disentangled graph super-network and the self-supervised training with joint architecture-graph disentanglement are novel contributions that enhance the understanding and performance of the proposed model.

- The extensive experimental evaluation on 11 real-world datasets demonstrates the superiority of the DSGAS model over several baseline methods, showing its effectiveness in unsupervised graph neural architecture search.


**Weaknesses:**

- While the paper presents a novel approach, further details regarding the implementation of the proposed DSGAS model in the main paper would be beneficial for the readers to understand the design clearly. For example, it is better to illustrate the details of the supernet construction in the main paper.

- Some typos should be corrected, e.g., heterogenous -> heterogeneous on line 323.


**Questions:**

Please see Weaknesses.

---

> ### Author Rebuttal · Authors · 2023-08-09
>
> We sincerely thank the reviewer for the valuable comments. We respond to the reviewer’s comments point by point as follows.
>
> > While the paper presents a novel approach, further details regarding the implementation of the proposed DSGAS model in the main paper would be beneficial for the readers to understand the design clearly. For example, it is better to illustrate the details of the supernet construction in the main paper.
>
> Thank you for your suggestion. We described the details of the super-network configuration in the appendix due to the page limit. Here, we briefly describe the super-network. The super-network consists of two parts, the operation pool and the directed acyclic graph (DAG). The operation pool includes several node aggregation operations (e.g., GCN, GIN, GAT, etc.), graph pooling operations (e.g., SortPool, AttentionPool, etc.), and layer merging operations (e.g., MaxMerge, ConcatMerge, etc.). The DAG determines how the operations are connected to calculate the graph representations for the subsequent classification tasks. We will add the illustrations in the revised main paper.
>
> > Some typos should be corrected, e.g., heterogenous -> heterogeneous on line 323.
>
> Thank you for your suggestions. We will correct the typos in the revised main paper.

---

> > ### Comment · Reviewer_GehM · 2023-08-18
> >
> > Thank you for the response. The rebuttal resolved my concern. In my opinion, the paper  addresses an important and underexplored nas problem, and the proposed disentangled self-supervision is novel. I would like to raise my recommendation from 7 to 8.

---

### Official Review · Reviewer_Hgo9 · 2023-07-07

**Soundness:** 3 good
**Presentation:** 3 good
**Contribution:** 3 good
**Rating:** 7
**Confidence:** 5

**Summary:**

This paper gives a pioneer solution for graph neural architecture search with limited labels. The key idea is to train a super-network containing disentangled factor-wise architectures by a self-supervised learning specially designed for graph neural architecture search. In this way, the paper addresses the key problem of finding the optimal architectures capturing various graph factors in an unsupervised fashion. The extensive experiments show that the method achieves signifiant improvements over the state-of-the-art GraphNAS methods under unsupervised settings. Detailed ablation studies also show the effectiveness of each proposed components.


**Strengths:**

Unsupervised GraphNAS is an important and valuable problem which remains unexplored in the literature. This paper is a timely work to introduce GraphNAS in the scenarios where labels are scarce, and these scenarios are actually quite common in practice. The method design is novel, where the contrastive search module is especially interesting that it pushes together architectures with similar factor expertises and pull away unsimilar architectures by architecture augmentations to explore the search space. The extensive experiments verify its capability of automating the architecture design for the data from various fields.


**Weaknesses:**


1. Although it may fall outside the scope of this paper, i'm curious about whether the model can be applied into areas like AI4Science where labels are quite scarce?

2. Pretraining techniques have shown promising applications in many scenarios like NLP and CV, and they usually have a close relationship with unsupervised or self-supervised techniques. It would better to discuss the relationship between this method and GraphNAS pretraining.

3. The framework diagram (figure 1) is a bit small, especially that the font size which is too small to read clearly. It would be better to enlarge the diagram and increase the font size a bit.


**Questions:**


1. Although it may fall outside the scope of this paper, i'm curious about whether the model can be applied into areas like AI4Science where labels are quite scarce?

2. Pretraining techniques have shown promising applications in many scenarios like NLP and CV, and they usually have a close relationship with unsupervised or self-supervised techniques. It would better to discuss the relationship between this method and GraphNAS pretraining.


**Limitations:**

I don't see any concerns regarding the societal impact in this work.

---

> ### Author Rebuttal · Authors · 2023-08-04
>
> We sincerely thank the reviewer for the valuable suggestions, which are helpful for the improvement of the paper. We respond to the reviewer’s comments point by point as follows.
>
> > Although it may fall outside the scope of this paper, i'm curious about whether the model can be applied into areas like AI4Science where labels are quite scarce?
>
> Thanks for your inspiring comment. We agree that there exist some trending areas like AI for Science that are close to the problem we focus on, i.e., graph neural architecture search with limited or even no labels. For instance, we notice that some recent works attempt to comprehend the relationship between metabolic pathways and molecular pathways for synthesizing new molecules by leveraging the power of graph neural networks. In these scenarios, labels, e.g., the molecular properties or subtypes, are limited, posing great challenges for graph neural architecture search to automatically enhance the GNN architectures. Since our method is specially designed for GNAS with limited or even no labels, these scenarios are natural and promising applications, which we leave to future works for explorations.
>
> > Pretraining techniques have shown promising applications in many scenarios like NLP and CV, and they usually have a close relationship with unsupervised or self-supervised techniques. It would better to discuss the relationship between this method and GraphNAS pretraining.
>
> Thanks for your comment. To the best of our knowledge, it remains unexplored for GraphNAS pretraining, which could simultaneously contain multiple goals like self-supervised learning, transfer learning, multi-task learning, etc. In this paper, we mainly focus on unsupervised graph neural architecture search, i.e., to discover graph architectures without labels. In semi-supervised experiments, we also show some progress in pretraining the super-networks to alleviate the label scarcity issues. For example, our model with the pretraining stage has an absolute improvement of 5% on one dataset compared with the ablated version without the pretraining stage. We will leave exploring more aspects of GraphNAS pretraining in future works.
>
> > The framework diagram (figure 1) is a bit small, especially that the font size which is too small to read clearly. It would be better to enlarge the diagram and increase the font size a bit.
>
> Thank you for your suggestions. We will improve the diagram in the revised main paper.

---

### Official Review · Reviewer_5ige · 2023-07-08

**Soundness:** 3 good
**Presentation:** 3 good
**Contribution:** 3 good
**Rating:** 6
**Confidence:** 4

**Summary:**

This paper mainly focuses on the problem of graph neural architecture search without labels. The authors find that the key problem is to discover the latent graph factors that drive the formation of graph data as well as the underlying relations between the factors and the optimal neural architectures. To this end, the authors propose the disentangled super-network, and design a self-supervised training and searching paradigm to automate the architecture design. The experiment results are significant and the deeper analyses are convincing.

**Strengths:**

- This paper mainly focuses on the problem of graph neural architecture search without labels, which is important yet underexplored in the literature.

- The proposed methods sound reasonable.

- The experiment results are significant and the deeper analyses are convincing.

**Weaknesses:**

- Can the authors explain the details of the intuition for the proposed contrastive search?

- The experiments include both node classification and graph classification tasks. Do these tasks share the same search space?

- As the existing graph NAS methods are in need of labels for training and searching, how do the authors modify them to compare as baselines?

- What about the complexity of the method?

**Questions:**

Please see the weaknesses.

**Limitations:**

None.

---

> ### Author Rebuttal · Authors · 2023-08-09
>
> We sincerely thank the reviewer for the detailed comments and suggestions. We respond to the reviewer’s comments point by point as follows.
>
> > Can the authors explain the details of the intuition for the proposed contrastive search?
>
> Thank you for your suggestion. We provide the details of the intuition for the proposed contrastive search as follows. As architectures similar in operation choices and topologies have similar capabilities of capturing semantics for downstream tasks, slightly modifying the architecture will have a slight influence on its capability. Moreover, since different GNN architectures are experts in different downstream tasks, the architectures searched for different disentangled latent factors are expected to have dissimilar capabilities under different factors. For these two reasons, we propose the contrastive architecture search to capture discriminative features by pulling similar architectures together and pushing dissimilar architectures away in the latent space.
>
> > The experiments include both node classification and graph classification tasks. Do these tasks share the same search space?
>
> Thank you for your comment. In comparison with the search space for node classification tasks, the search space for graph classification tasks has two extra kinds of tailorable operations, i.e., graph pooling operations to capture global representations and layer merging operations to provide jumping knowledge. We provided the details of the search space in Appendix D.1.
>
> > As the existing graph NAS methods are in need of labels for training and searching, how do the authors modify them to compare as baselines?
>
> Thank you for your question. We first pre-train the models by self-supervised loss and then evaluate the models by finetuning an extra classifier. For the graph NAS baselines, the self-supervised loss is utilized to substitute the supervised loss to train the model parameters as well as select the architectures.
>
> > What about the complexity of the method?
>
> Thank you for your suggestion. We provide the complexity analysis as follows. Denote the number of nodes and edges in the graph as $N$ and $E$, the number of latent factors as $K$, the number of operation choices as $|\mathcal{O}|$, the dimensionality of hidden representations as $d$. The time complexity of the disentangled super-network is $O(K|E|d + K|V|d^2)$, where the computation for each factor is fully parallelizable and amenable to GPU acceleration, and $K$ is usually a small constant. The time complexity of the self-supervised training and contrastive search modules is both $O(K^2d^2)$. As architectures under different factors share the parameters, the number of learnable parameters is the same as classical graph super-network, i.e., $O(|\mathcal{O}|d^2)$. Therefore, the complexity of our method is comparable to classical GNAS methods.

---

### Official Review · Reviewer_bqLE · 2023-07-08

**Soundness:** 3 good
**Presentation:** 2 fair
**Contribution:** 3 good
**Rating:** 7
**Confidence:** 5

**Summary:**

In this paper, the authors study the problem of unsupervised graph neural architecture search, which remains unexplored in the literature. The authors propose a novel Disentangled Self-supervised Graph Neural Architecture Search (DSGAS) model, which is able to discover the optimal architectures capturing various latent graph factors in a self-supervised fashion based on unlabeled graph data. Extensive experiments on 11 real-world datasets demonstrate that the proposed DSGAS model is able to achieve state-of-the-art performance against several baseline methods in an unsupervised manner.

**Strengths:**

1. I believe the problem of unsupervised graph neural architecture search is important, and this work is significant that it may unlock more NAS applications in practice.

2. The writing is clear and well-organized in general.

3. The experments and analyses are extensive.

**Weaknesses:**

Please see my comments in the questions part.

**Questions:**

1. What is the relationship between supervised and unsupervised nas paradigms in terms of the optimization problem, e.g., line 84 in the main paper?

2. In Figure 1, the super-network seems a simple DAG, why do the searched architectures in Figure 4 seem kind of complex in terms of the operation connections?

3. In Figure 1, it seems that the mixed operation has different colors for different $\alpha_i$, is it possible that some factor-wise architectures share the same operation?

4. In Figure 1, is it required that all the architectures should choose the same augmentations? In other words, is it possible that some architectures adopt operation choice perturbation, while others adopt weight perturbation in the same batch for contrastive learning?

---

> ### Author Rebuttal · Authors · 2023-08-04
>
> We sincerely thank the reviewer for the detailed comments and insightful questions. We make responses to the reviewer’s comments as follows.
>
> > What is the relationship between supervised and unsupervised nas paradigms in terms of the optimization problem, e.g., line 84 in the main paper?
>
> Thank you for your comment. The problem of unsupervised GNAS can be formulated as optimizing an architecture generator that is able to discover powerful architectures by exploiting inherent graph properties without labels. Since the labels are not available under unsupervised settings, the validation and training metrics in Line 84 can not be calculated to measure the architecture performance and search architectures as in supervised NAS methods.
>
> > In Figure 1, the super-network seems a simple DAG, why do the searched architectures in Figure 4 seem kind of complex in terms of the operation connections?
>
> Thank you for your question. The proposed disentangled super-network can flexibly incorporate multiple architectures to be searched with regard to various graph factors. This design alleviates the entanglement of architectures by providing more flexible choices of paths in the super-network, which also results in complex yet competitive architectures.
>
> > In Figure 1, it seems that the mixed operation has different colors for different $\alpha_i$, is it possible that some factor-wise architectures share the same operation?
>
> Thank you for your question. Yes, some factor-wise architectures can share the same operation.  During the search process, the architectures are encouraged to capture different graph factors but are not prohibited from sharing the same operation. This enables the factor-wise architectures to share some common knowledge in learning the graph properties.
>
> > In Figure 1, is it required that all the architectures should choose the same augmentations?
>
> Thank you for your question. No, the architectures can choose different augmentations. Like the data augmentations in contrastive learning, all the architecture argumentations act as a transformation of an architecture to a similar architecture. In Figure 3(b), 'Compose' randomly chooses different architecture augmentations for each architecture, and it is shown to be effective on various datasets.

---

> > ### Comment · Reviewer_bqLE · 2023-08-19
> > **Thanks for your reply.**
> >
> > Thanks for your detailed response. All my concerns have been resolved. Besides, after reading other reviews and your reply, I found the NAS disentangled mechanism is pretty interesting and sounds good. I would like to raise my score.

---

### Decision · Program_Chairs · 2023-09-21

**Decision:**

Accept (poster)

**Comment:**

This work proposes a novel, disentanglement-based approach to graph neural architecture search. All reviewers assess that the work is a significant contribution, with very interesting insight---many of the reviewers strongly argued for acceptance. While some concerns were raised about the method's motivation and reproducibility, the Authors provided a convincing rebuttal that clears the residual doubt. I strongly support this work for acceptance at NeurIPS.